# Technologies and Data Analytics to Manage Grain Quality On-Farm—A Review

Cassandra K. Walker [1,2,*], Sahand Assadzadeh [1], Ashley J. Wallace [1], Audrey J. Delahunty [1], Alexander B. Clancy [1], Linda S. McDonald [1], Glenn J. Fitzgerald [1,2], James G. Nuttall [1,2] and Joe F. Panozzo [1,2]

[1] Agriculture Victoria Research, Grains Innovation Park, Horsham, VIC 3400, Australia; sahand92@gmail.com (S.A.); ashley.wallace@agriculture.vic.gov.au (A.J.W.); audrey.j.delahunty@agriculture.vic.gov.au (A.J.D.); alex.clancy@bayer.com (A.B.C.); linda.mcdonald@agriculture.vic.gov.au (L.S.M.); glenn.fitzgerald@agriculture.vic.gov.au (G.J.F.); james.nuttall@agriculture.vic.gov.au (J.G.N.); jpanozzo@unimelb.edu.au (J.F.P.)

[2] Centre for Agricultural Innovation, School of Agriculture and Food, Faculty of Science, The University of Melbourne, Parkville, VIC 3010, Australia

* Correspondence: cassandra.walker@agriculture.vic.gov.au

**Abstract:** Grains intended for human consumption or feedstock are typically high-value commodities that are marketed based on either their visual characteristics or compositional properties. The combination of visual traits, chemical composition and contaminants is generally referred to as grain quality. Currently, the market value of grain is quantified at the point of receival, using trading standards defined in terms of visual criteria of the bulk grain and chemical constituency. The risk for the grower is that grain prices can fluctuate throughout the year depending on world production, quality variation and market needs. The assessment of grain quality and market value on-farm, rather than post-farm gate, may identify high- and low-quality grain and inform a fair price for growers. The economic benefits include delivering grain that meets specifications maximizing the aggregate price, increasing traceability across the supply chain from grower to consumer and identifying greater suitability of differentiated products for high-value niche markets, such as high protein product ideal for plant-based proteins. This review focuses on developments that quantify grain quality with a range of spectral sensors in an on-farm setting. If the application of sensor technologies were expanded and adopted on-farm, growers could identify the impact and manage the harvesting operation to meet a range of quality targets and provide an economic advantage to the farming enterprise.

**Keywords:** in-field; spectroscopy; image analysis; machine learning; protein; grain size; grain color

## 1. Introduction

The application of sensor technologies in agriculture is having an increasing role in measuring grain yield potential and grain quality throughout the cropping season. Important global arable crops include wheat, maize, canola, rice, soybean and barley due to their nutritional importance, functional properties, and commodity value. The grain quality of commercially grown crops is influenced by a range of factors, including cultivation practices, environment, harvest timing, grain handling, storage management and transportation [1–7]. The spatial variation observed in grain yields can be due to soil type, topography and interactions with the environment (e.g., frost and water availability) within and across fields and farms and is well recognized as a driver of variation in grain quality. In addition, with the increasing likelihood of extreme events and biotic pressures associated with climate change, it will be more challenging to maintain grain quality in future environments. Although growers currently manage these variations through better-adaptive cultivars, management practices and in-field sensors could provide

improvements in managing grain quality and, ultimately, profits by providing information that would allow identification and segregation of grain quality by quantifying variation in quality obtained from the field. This would benefit end-users and add value to the farming enterprise.

Grain quality is defined by a range of physical and compositional properties, where the end-use dictates the grain and compositional traits and market potential [8–10]. Grain quality has traditionally been measured post-farmgate when growers deliver to grain receival agents, and the load is subsampled and tested within a clean testing environment such as a testing station or laboratory using benchtop instrumentation and human visual inspection. Grain quality when received by grain receival agents is graded using two approaches: firstly, subjective procedures undertaken by trained operators (grain inspectors) assessing visual traits, including stained, cracked, defective grain and contaminants, and secondly, objectively using standardized instrumentation, such as sieves to determine grain size and near infrared spectroscopy (NIR) to determine the composition, such as protein and moisture concentration [10,11]. Key traits used in valuing and trading grain include the percentage of small grain (screenings), foreign material (unable to be processed, milled or malted), contaminates, sprouted, stained or discolored grain, broken, damaged or distorted grain, presence of insects or mold, test weight, or composition including moisture concentration, protein concentration, low levels of aflatoxins and a high Hagberg falling number test [9,12–14]. Many of these traits are measured as % per weight of samples subsampled from the grain load. Manufacturers of end-product determine the specifications or limits associated with these quality traits, as grain outside the set specifications impacts the end-product quality. For example, small, shriveled grain that may be high in protein concentration would be undesirable when processing food products from wheat as these impact flour yields, baking loaf volume and dough rheology characteristics as reviewed by [15].

The application of sensor technologies pre-harvest is increasingly being incorporated into farming enterprises to determine the impact of environmental factors on grain production and yield. For example, field sensors are used to discern spatial and temporal information throughout the production system, including soil type variation and nitrogen inputs (e.g., electromagnetic conductivity [EM38]), growth habit (Normalized Difference Vegetative Index [NDVI] and relative greenness [SPAD]) [16,17], nutritional status (Canopy Chlorophyll Content Index [CCCI]) [18], the impact of abiotic stresses such as frost and heat [4,19], and biotics where RGB (Red, Green, Blue) imaging is used to target spraying weeds [20]. The next transition is to utilize sensors to record reliable, efficient, and relevant data associated with grain quality. Downgrades in grain quality can be caused by heat waves and frost during grain filling, drought, disease and the presence of weeds or other contaminants. Previous studies have shown that stresses such as high temperature, chilling, water stress (associated with soil type), and disease can vary spatially within grower's fields [21–24]. Studies linking the spatial distribution of stress events to grain quality have predominantly been related to protein and protein quality in wheat [25,26]. However, limitations remain as to how sensor technologies can be applied to all crop types for a range of grain quality traits.

The monitoring of grain quality, pre- or post-harvest, is currently limited by the costs of the sensors, their availability, and the development of algorithms for measuring key grain quality traits. A deeper understanding of the grower's perspective, in particular, the key quality traits of relevance to their farm business, is required to guide priorities when developing these applications. Consideration must also be given to a range of practical issues associated with deploying sensors on-farm and managing the logistics of a segregated grain supply chain. Previous studies have highlighted some opportunities to improve grower returns through segregating cereals at harvest [2,27–29]; the economics of actively managing grain quality on-farm needs to be quantified at the practical level.

Given the speed of sensor development and miniaturization, sensor technologies are commercially available that produce reliable data in an on-farm environment rather than

in a clean laboratory. Moving sensors on-farm and in-field enable growers to strategically manage grain quality before the product is sold (post-farmgate) (Figure 1). The most common commercially available sensor that determines grain composition is the benchtop NIR. Growers subsample harvested grain and use these systems to determine moisture and protein prior to sale or storage.

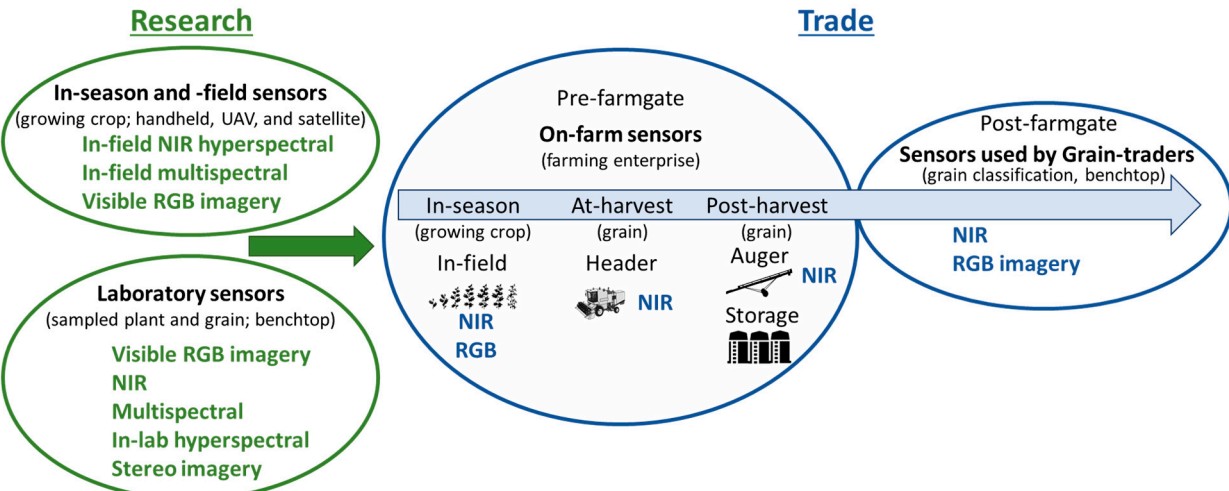

**Figure 1.** Commercially available practices used to predict grain quality in research and for trade purposes.

Technologies that are commercially available that have the potential to objectively measure grain quality for trade purposes and are likely to be adopted within the next 5 to 10 years include NIR, digital RGB imaging, multi- and hyper-spectral RGB and NIR sensors (Figure 1). These sensors enable rapid and non-destructive approaches to determine grain quality and are used to analyze plant and grain material for research purposes in-season, at-harvest, and post-harvest [19,22,30–32]. In this review, the focus is on digital imaging and NIR spectral technologies, as these are currently used in research in-lab and, as described, are moving from the lab to the farm, but there are several considerations needed to make this possible. The focus is the commercial and research landscape of current and emerging visible and NIR sensor systems, with consideration of the practicality of applying these technologies on-farm to assess quality traits used to determine market value. Low-cost portable sensors are moving from research, benchtop and laboratory applications to on-farm practical applications where grain quality can be measured objectively in real-time and inform grain management decisions prior to sale post-farm gate.

A critical step in applying sensor technology for determining grain quality is the analysis of the data collected. The application of different analytical techniques is dependent on the types of data generated by different types of sensors. Color space analysis [33], for example, is appropriate to use with digital imagery (RGB), while machine learning methods generally require very large data sets and are adept at finding patterns within high-dimensional data sets. As processing speeds increase and computational costs decrease, even more data-intense analytical techniques can be used to help interpret data in near real-time. To make the best use of sensors and the increasingly large amounts of data produced, it in critical to select appropriate software analytical techniques to work hand in hand with the sensor hardware.

This review summarizes current sensor technologies that are in the research phase and can potentially be applied on-farm within the next 5 to 10 years, algorithm approaches that can be applied and the planning, logistics and decision-making processes involved with determining grain quality on-farm to classify market value.

*1.1. Options for On-Farm Sensing Grain Quality*

Within dryland cropping systems, proximal and remote sensing technologies are being used to capture and manage temporal and spatial variability that cause variation in crop growth, yield and quality across the landscape [34,35]. Remote sensing can operate at multiple scales to assess crop growth and stress, including using hand-held instruments at the field scale, airborne platforms at the field and farm scales and satellites at scales from field to regional levels (Figure 1). For example, a crop reflectance index, referred to as the Normalized Difference Vegetation Index (NDVI), measures the difference between near infrared (vegetation strongly reflects) and red light (vegetation absorbs). This index measures crop canopy cover and can correlate with plant biomass enabling the monitoring of vegetation systems [36]. In one long-term study which assessed the response of winter wheat to heat stress on the North China Plain, canopy reflectance measured from satellite platforms [37] was used to assess crop phenology and senescence rate (spatially and temporally), providing insight into climate change impacts on broadscale production potential. Canopy reflectance information using targeted spectral indices from the visible and near infrared spectral regions has also been correlated with plant nitrogen status in wheat [38], leading to the development of indices including the Canopy Content Chlorophyll Index (CCCI) and the Canopy Nitrogen Index (CNI) at the experimental plot scale. Further testing of these indices across different dryland growing environments in Australia and Italy confirmed their utility, with good agreement between these indices and canopy nitrogen concentration ($r^2 = 0.97$) early in the growing season [18]. In this example, such canopy-based remotely sensed indices provide growers with a decision support tool to guide in-season nitrogen fertilizer application, enabling spatial management of fertilizer rate across the paddock, thus optimizing the match between plant nitrogen demand and application.

More recently, crop canopy reflectance characteristics have also been used to monitor frost damage in wheat grown in southern Australia [19,22], where rapid estimation of frost damage to crops on a spatial basis supports the timely management decisions by growers to reduce the economic impact of frost. In these studies, where hyperspectral reflectance and active light fluorescence were assessed, the reflectance indices Photochemical Response Index (PRI), Normalized Difference Red-Edge Index (NDRE), NDVI and the fluorescence-based index FLAV (Near Infrared Fluorescence excited with Red/Infrared Fluorescence excited with UV (375 nm) correlated well with frost damage experimentally imposed to field grown wheat at flowering. These principles are now evolving to a more widespread application within the grains industry, with service providers now offering seasonal satellite imagery and interpretation in the context of crop performance, spatial variation (topography) and soil characteristics. These data can be useful to inform the management of distinct parts of a paddock or farm (as management zones) for nutritional, abiotic, and biotic disorders to maximize yields and ensure high grain quality.

Research undertaken on-farm relates monitoring of canopy reflectance to crop growth and yield. However, this review highlights the importance and value of predicting the quality of the end-product, i.e., the grain, for growers on-farm. To predict grain-yield, a study that employed hand-held hyperspectral devices assessed variation in frost-affected lentil crops and confirmed that frost damage could be effectively detected in-season using remote sensing and that the stress response corresponds to both yield and quality outcomes [4,39]. Similarly, the use of the active light fluorometer index FLAV in chickpeas and the spectral reflectance-based Anthocyanin Release Index 1 in faba beans have shown utility for the detection of the disease Ascochyta blight at the leaf-level [40]. Given the potential for Ascochyta blight, a fungal disease in pulse grains caused by *Ascochyta rabei*, to affect grain quality, this offers the opportunity for improved control of diseases through spatial management of foliar fungicides [40], potentially reducing associated quality downgrades and seed carryover. More broadly, if variation in crop canopy reflectance (NIR) based on environmental stresses could be linked to grain quality, this would allow in-season mapping to predict spatial variation in grain quality prior to harvest. With this knowl-

edge, the harvesting program could be tactically designed to harvest zones based on predicted yield and limit economic losses to environmental effects on grain quality within the grains industry.

Monitoring crop canopies using spectral analysis has the potential to provide growers with the tools to better manage grain quality. For this technology to be applied on-farm, further research is required to identify remote sensing measures that correlate with grain quality outcomes for key crop species. The logistics of employing such techniques on-farm must also be considered; the timeliness of collecting and interpreting spatial data to inform harvest zoning and the physical infrastructure and logistics to accommodate the various quality grades are two such examples. Developing prediction models for grain quality based on sensor data will inform harvest zones for quality and maximize financial return.

*1.2. Data Acquisition Systems and Sample Handling*

In addition to monitoring and managing grain quality prior to harvest, there are opportunities to directly monitor grain quality at various points in the supply chain from harvest through to sale. In this case, sensors could be installed within harvesting or grain handling machinery [26], in grain storage infrastructure [41], or in a 'benchtop' or laboratory setting similar to grain receival points [32]. Compared with in-season quality monitoring, assessment of grain quality at harvest or during movement to storage offers the advantage of avoiding the need for mapping and interpreting quality data prior to harvest, saving an additional operation. It may also allow for the identification of quality defects not otherwise evident through spectral analysis of the crop canopy; for example, instances where weeds or other contaminants (e.g., snails) arise closer to crop maturity.

Monitoring of grain quality at- or post-harvest can also be undertaken spatially, enabling proactive management in future seasons. Commercially available grain protein monitors are one such example [26,42], enabling segregation based on grain quality at harvest or during movement into or out of storage while providing information to inform future decisions relating to nitrogen fertilizer. In addition to segregating grain based on quality grade, monitoring quality at- or post-harvest may enable growers to blend grain of various quality levels in specific proportions prior to entering storage [42]. This strategy can be used to 'lift' low-quality grain from one grade to the next, helping to achieve the highest aggregate quality and price for grain produced across a farm business.

In addition to monitoring grain quality at-harvest or during movement into storage, directly monitoring grain quality could also be undertaken during storage. Studies such as [43,44] have shown that grain quality can change over time. While optimizing storage conditions through aeration, temperature control and fumigation can all help to maintain grain quality for longer, monitoring of stored grain in situ offers the potential to target marketing decisions based on projected commodity prices and prediction of grain quality over time.

In considering the on-farm application of grain quality monitoring systems, it is important to understand accuracy requirements. The ultimate success of quality monitoring is dependent on its ability to correctly classify a known volume of grain into its respective quality grade to trade. The accuracy required will depend on the crop species and the quality grade being targeted. This is further complicated where the quality of a given grain parcel varies across different target traits, e.g., contamination, color, and size, requiring more comprehensive monitoring systems with multiple algorithms (and perhaps different technologies) to account for the different traits within the grade. Typically, the utility of a monitoring system depends on the accuracy of both the instrument and the sampling process. Analyzing a sample that is representative of a given parcel of grain is vital to ensure the accuracy of the monitoring system [45] and the ability to apply it in an on-farm situation.

## 2. Sensor Technologies Used in the Agricultural Production System

Key sensor technologies employed post-farmgate to objectively measure grain quality include RGB imagery, NIR spectroscopy and multispectral spectroscopy. Hyperspectral spectroscopy will become more widely adopted for commercial applications on farms as new cheaper sensors are developed and manufactured. These technologies and grain quality applications are summarized in Table 1, and their advantages and limitations are outlined below.

**Table 1.** List of sensor technologies and examples of their applications in evaluating grain quality associated with measuring traits important for on-farm applications.

| Sensor Type | Spectral Range (nm) | Application | Product Stage | References |
|---|---|---|---|---|
| Digital camera (benchtop) | RGB [1] | separation of red vs. white wheat, Fusarium damaged vs. undamaged, high vs. medium vs. low protein | Research | [46] |
| | | measure grain color, % hard endosperm | Patent | [47] |
| | | gradation of color, whiteness, and hard endosperm of the seed/grain | Patent | [48] |
| | | measure grain plumpness, density, and volume | Research | [49,50] |
| Stereo camera (in-field, mounted cameras) | RGB | crop height estimation | Research | [51] |
| | | wheat canopy structure | Research | [52] |
| Near infrared (benchtop) | 900–1600 | variety identification and seed health | Research | [53] |
| | 950–1650 | separation of red vs. white kernels, Fusarium damaged vs. undamaged, high vs. medium vs. low protein | Research | [46] |
| | 1100–2300 | prediction of protein and moisture constituency and grain type classification | Research | [31] |
| | 850–2300 | moisture, protein, oil concentration | Commercial | [54] |
| Multispectral (benchtop) | 360–950; 8 bands | characterization of desiccation | Research | [55] |
| | 375–970; 19 bands | variety identification and seed health | Research | [53] |
| | 375–970; 19 bands | seed authentication; % adulterated samples | Research | [56] |
| | 360–950; 8 bands | fungal contamination detection | Research | [57] |
| | 360–950; 8 bands | identification of the histological origin | Research | [58] |
| Hyperspectral (in-field point-based) | 405–850; 6 bands | assessment of lentil size traits | Research | [32] |
| | 405–850; 6 bands | classification of defective vs. non-defective field pea | Research | [13] |
| | 400–1000 350–2500 | detection and classification of biotic and abiotic stresses in-field; crop yield predictions and mapping | Research | [19,22,24,39,40] |
| Hyperspectral (benchtop) | 400–1000 | seed authentication | Research | [56] |
| | 400–1000 | detection of green vs. normal barley kernels | Research | [59] |
| | 980–2500 | protein in single wheat kernels | Research | [60] |
| | 1000–2500 | sprouting, enzymatic activity | Research | [61] |
| | 375–1050 | micronutrient composition; Ca, Mg, Mo, and Zn in wheat | Research | [62] |

[1] RGB = Red, Green, Blue (nominally ~400–700 nm).

*2.1. Digital RGB Camera*

Digital cameras are one of the most widely used sensors with broad applications ranging from industrial quality control and robotics to capturing photographs of scenes and objects. These sensors can provide data-rich information and are used to analyze a wide range of visual traits, key classifying grain quality [63]. Digital (RGB) cameras are typically equipped with light-sensitive Complementary Metal–Oxide–Semiconductor (CMOS) or Charge-Coupled Device (CCD) sensors to acquire colored images of scenes and objects [32,63]. Color filter arrays (CFAs) arranged in a mosaic pattern on top of the sensor selectively filter RGB light, and the intensity of the light in each color channel at different spatial points on the sensor is measured and used to compose an RGB image [63]. RGB cameras are commercially sold as DSLR (Digital Single Lens Reflex), small-scale machine vision cameras, industrial cameras, or as a device component, for example, in smartphones. The specific application of a camera depends on the resolution of the captured images, optics, light and color sensitivity, and frame rate, among a range of other factors. Factors including the signal-to-noise ratio and size of the sensor, optics, illumination and processing of the image (e.g., interpolation of pixels) determine the raw image data. The resolution of a static photo is usually expressed in megapixels, i.e., the number of pixels (length × width) in the image.

Digital cameras equipped with CMOS sensors are used for process-control and monitoring automated systems. Their adoption is mostly driven by reduced costs, improved power efficiency of the CMOS chips, and advancements in machine vision techniques for the analysis of the captured data. CMOS cameras can operate with both passive (sunlight) and active lighting, be deployed on vehicles or UAVs for remote monitoring, and the relatively small size of the generated data (e.g., compared to in-laboratory hyperspectral imaging) enables images to be processed in real-time.

Computer Vision (CV) is the scientific field in which computational methods are developed to derive meaningful information from camera data (images and videos). Computer vision includes computational methods for object and motion tracking, scene understanding, edge detection, segmentation, color and texture analysis [64], object detection, classification, and counting, among some of the applications [65].

Many indicators of food and plant quality, including ripeness of fruit, symptoms of diseases, nutrient deficiencies, damaged plants, weeds and plant species in crop fields, and others, manifest visually and thus assessed according to visual criteria; these features are suitable targets for detection by RGB cameras [32,63,66–70]. Throughout the agricultural industry, the assessment of morphological and phenotypic features usually requires a visual inspection, a labor-intensive process prone to subjectivity and errors. Digital image analysis can improve methods of grain and plant classification according to objective visual criteria. Applications of RGB imaging in grain quality are outlined in Table 1.

The advantage of using RGB cameras is that images are readily interpretable because the information captured is how a human perceives the appearance of the grain sample. Regarding the limitations of applying machine vision, these are similar to those encountered in human vision in that they both operate within the visible range of the light spectrum. Quantifying color using RGB imaging systems is complex as it can be difficult to correct or calibrate between systems due to differences in the camera optics, sample illumination, consistent presentation of the sample for image capture and different ways images can be processed and compressed for storage. The use of consistent lighting across samples and the inclusion of calibration panels within images is critical to robustly quantify grain characteristics. Digital imaging captures the external view of samples, where the internal grain structure and chemical information is not detected in RGB images. Therefore, to analyze sub-surface grain features, other spectroscopic techniques would need to be engaged, including NIR, Raman, NMR, UV, X-ray and fluorescence. Another limitation is the sample orientation. RGB imaging captures one viewpoint of a sample; therefore, traits, such as surface area, volume, length, width and diameter of samples, may not be representative depending on sample presentation. Also, certain complex processing traits

have been analyzed using image analysis, such as grain milling, the error can be high and the range in laboratory values limited, compromising the accuracy and precision of the technique [30,71].

### 2.2. Stereo Cameras

Stereo vision and Structure from Motion (SfM) are two methods used for three-dimensional (3D) imaging with conventional digital cameras [72]. Binocular stereo vision uses images captured from two cameras at different angles and triangulation to compute the depth of scenes, analogous to the human binocular depth perception. In SfM, a single moving camera is used to obtain multiple images from different locations and angles, and the images are processed to obtain the 3D information. Stereo vision enables the estimation of the volume of objects and their spatial arrangements. Most of the applications of stereo cameras in agriculture are field-based and have been to generate 3D field maps [73] for biomass estimation [74], determine morphological features of wheat crops [52] and crop status, including growth, height, shape, nutrition and health [72]. There are limited applications of SfM for grain quality analysis; these include stereo imaging systems with two viewpoints, which have enabled the prediction of grain length, width, thickness and crease depth in wheat [75].

As with the two-dimensional (2D) or single imaging RGB systems, similar limitations are found with stereo vision systems: the information collected is within the visible range (RGB), sample color, differences between imaging systems, sample lighting, internal grain structure and chemical information is not detected, and sample orientation may not be visible. The benefit of using a stereo imaging system is that multiple positions of the sample are captured, and, therefore, the data is not limited to one viewpoint, allowing a complete view of the sample and thereby reducing the error in estimating traits such as surface area and sample volume.

### 2.3. Near Infrared Spectroscopy

The near infrared (NIR) region of the electromagnetic spectrum encompasses the 700–2500 nm wavelength range and covers the overtone and combination bands involving the C–H, O–H, and N–H functional groups, all of which are prevalent in organic molecules [54,76]. NIR spectroscopy is widely applied to the analysis of materials in agriculture, biomedicine, pharmaceutics, and petrochemistry. In the agriculture industry, NIR spectroscopy is used for determining the physicochemical properties of forages, grains and grain products, oilseeds, coffee, fruits and vegetables, meat and dairy, among many other agricultural products [54,76].

The instrumentation used in NIR spectroscopy consists of a light source (typically an incandescent lamp with broadband NIR radiation), a dispersive element (commonly a diffraction grating) to produce monochromatic light at different wavelengths, and a detector that records the intensity of the reflected or transmitted light at each wavelength [76]. An alternative method of obtaining the same information is with Fourier Transform near infrared (FT-NIR) spectroscopy, where the spectrum is reconstructed from interference patterns produced by a Michelson interferometer (interferogram) within the instrument [77]. Compared to dispersive NIR, FT-NIR systems can collect spectra at higher spectral resolutions; however, unlike gases, this advantage is not significant in the analysis of liquids and solid samples, where the spectral bands are broad (>2 nm). For whole-grain analysis (e.g., protein and moisture constituency), the predictive performance of both types of instruments (FT and dispersive) is similar, indicating no advantage of either method over the other [78].

NIR sensor readings are referenced with 'white' (reference) and 'dark' scans obtained from highly reflective (assumed to be 100% reflective) flat and homogenous materials such as fluoropolymers like Spectralon®, and dark current (no light) signals, respectively. The reference panel provides calibration to reflectance, which is the physical measure of light from the surface of an object, and the dark measurement quantifies sensor noise. The most

common referencing methods assume a linear response for sample reflectance R, given by $R = \left( I_{sample} - I_{dark} \right) / \left( I_{white} - I_{dark} \right)$, where I is the intensity of the signal measured by the sensor and the dark (noise) from the sensor is removed from the sample and reference. Alternatively, sample reflectance may assume a non-linear relation, in which case the reflectance is typically modeled with a higher order polynomial equation, calibrated using a set of reflectance standards (e.g., Spectralon® doped with graded amounts of carbon black) whose reflectance span the ~0–100% reflective range.

NIR spectroscopy is widely adopted, is relatively affordable ($US5000–$50,000) depending on the application and spectral sensitivity needed and is available as both desktop and portable low-power instruments. NIR can provide accurate measures of the chemical constituency of a sample, including the $w/w\%$ of nitrogen concentration (and thus protein), moisture, carbohydrates, and oils, among other organic compounds. Because the NIR radiation can penetrate a sample, it can be used to investigate its chemical composition. Furthermore, in densely packed bulk grain where objects overlap and occlude one another, unlike in image analysis, the NIR device is operable and not sensitive to the orientation and careful arrangement of the individual grains and can be used to measure whole grain sample properties.

Traditional NIR spectroscopy sensors, unlike digital imaging, capture the average spectrum of a sample of grain packed within a measuring cell. Digital images capture two-dimensional information, and within the image features can measure grain size distribution, whereas NIR spectroscopy, typically in homogenous samples, quantifies an average (of the analyzed sample) quantity of all the individual grains within the sensor field of view. Complexities in sample composition, mixing of spectral components (different parts of the grains, shadows, contaminants, etc.), and low concentration of analytes can limit the accuracy of traits measured with a NIR instrument.

Applications for NIR spectroscopy include rapid determination of oil, protein, starch, and moisture in a range of grains and their products [54], including forages and food products [79]. Other applications include the identification of wheat varieties and seed health [53], fungal contamination [46] and prediction of protein and moisture concentration [54,79]. NIR has been widely adopted to measure protein and moisture concentration which is then used to determine the value (grade of the grain) and processing quality of the grain, i.e., baking quality in wheat and malting quality in barley.

### 2.4. Multispectral Imaging

Multispectral imaging acquires reflectance data at (often narrow) discrete bands (up to about 20) spanning the ultraviolet (UV), visible, and near infrared (NIR) regions of the electromagnetic spectrum. In contrast, RGB color images only provide data at three (broad wavelength) channels (R, G, and B) within the visible spectrum. Regardless of the modality (RGB, multispectral, or in-lab hyperspectral), the data is organized in three-dimensional numerical arrays (i.e., data cubes) where the first two dimensions (X and Y) correspond to the spatial information, and the third dimension (λ) stores the spectral information. There are three main methods for the acquisition of data in spectral imaging systems, named after the sequence of data acquisition along each of the X, Y and λ directions: (i) point-scanning (whiskbroom), (ii) line-scanning (push-broom) and (iii) area scanning methods. Multispectral imaging has similar applications to RGB imaging; however, as the spectral bands can extend beyond the visible region, multispectral imagers have been used to identify wheat varieties, detect black point disease or fungal contamination [53,57], track desiccation of seeds [55], seed authentication [56] and identify the histological origin of wheat grain [58]. Point-scanning involves the acquisition of a complete spectrum at each spatial point (X, Y), and the data is stored in band-interleaved-by-pixel (BIP) format. Because the spectrum at each pixel is acquired one at a time, this system is typically used in microscopy (e.g., atomic force microscopy) where acquisition speed is not a priority (because the object is not moving). In line-scanners, data is recorded line by line (y, λ) as the target sample moves along the X-direction. The data is stored in band-interleaved-by-line

(BIL) format. This configuration is typically used in industrial scanners where samples are scanned during their movement on a conveyor under the imaging system. This is also typical when the sensor itself is moving across a stationary target, such as when deployed from an aircraft. In area-scanners, an entire 2D image is acquired at each λ, which results in a band-sequential (BSQ) data format. This method requires a rotating filter wheel or a tunable filter (e.g., Liquid crystal tenable filter, LCTF, or Acousto-optic tunable filters, AOTF) to target the wavelengths of interest at each scan and is generally not suitable for moving samples, unless movement is minimal with a high degree of overlap [76]. Other imaging systems use LEDs of different emission wavelengths (UV—NIR) to sequentially illuminate objects placed in a dark enclosure to capture greyscale images, which are then multiplexed along the λ direction to form the multispectral data. Variations in illumination (due to lighting geometry and setup), sensor sensitivity, imaging method, and environmental conditions (e.g., temperature and humidity) can affect the data quality acquired by spectral imaging systems, hence calibrations of these systems are very crucial and important for their function [30]. In remote sensing applications, the set of calibrations are often referred to as radiometric calibrations, which additionally account for the effects of altitude, weather, and other atmospheric conditions [80].

Low-frequency NIR wavelengths or UV have been used in multispectral imagers that can penetrate objects to capture information beyond the surface images of standard RGB cameras. Therefore, features 'invisible' to RGB imaging can be used to determine the chemical composition of samples, albeit with limited accuracy, depending on the number and frequency of the spectral bands. Multispectral systems are limited in their capacity to measure the chemical composition of samples effectively because only a limited number of wavebands tend to be utilized to ensure the instrument is low-cost. Hyperspectral imaging and spectroscopy methods are suited for this purpose. Furthermore, multispectral cameras are more expensive than digital RGB cameras.

### 2.5. Hyperspectral

### 2.5.1. In-Field Measurements

In-field hyperspectral sensors collect single-point data from the target, usually either in the visible-NIR (350–700 nm) or from visible to mid-IR (350–2500 nm), depending on the type of materials used for the sensors. Because these sensors collect single-point data, they average the spectral response across the measured area. In-field hyperspectral sensors are used to quantify crop conditions, for example, the impact of biotic and abiotic stresses on grain yield and quality [4,22,39,40]. The advantages are the collection of detailed spectral signatures to quantitatively assess crop/plant health and function. Disadvantages include the cost of the portable instrumentation (>$US100,000), the complexity of calibration, removing noisy data due to sky and environmental conditions, collection of appropriate and relevant reference data and the development of robust calibrations that can be applied across a range of weather conditions.

### 2.5.2. In-Laboratory Hyperspectral Imaging

Laboratory (in-lab) hyperspectral imaging systems combine the benefits of imaging and spectral systems to simultaneously acquire spectral and spatial information in one system. This can be applied to the quantitative prediction of the chemical and physical properties of a specimen as well as their spatial distribution simultaneously. The inner workings of in-lab hyperspectral systems are like multispectral systems, with variants that use dispersive optical elements or LCTF/AOTF to record spectral signals. However, unlike multispectral images where up to 20 discrete bands are recorded, a hyperspectral system typically acquires several hundred contiguous wavelength data points at each image pixel. The most common dispersive hyperspectral imaging sensor is the push-broom line scanner [76].

In-lab hyperspectral imaging has been used extensively for quality evaluation of fruits and vegetables, enabling the detection of contaminants [81], bruises [82], rot [83] and quality

attributes including firmness [84], moisture and soluble solid concentration (SSC) [85,86], and chilling damage [87]. The publication of applications using hyperspectral technologies in agriculture has increased over the past 30 years, with over 245 articles published from 2011 to 2020 [88], and relevant on-farm applications are outlined in Table 1.

The major limiting factors for the implementation of hyperspectral systems that measures within the NIR spectrum (980–2500 nm) are cost (~USD 200,000), high dimensionality, and volume of captured data, creating challenges in online real-time systems. It should be noted that hyperspectral imagers that measure the visible region or short wavelengths are considerably cheaper and, therefore, can be applied to specific applications. Hyperspectral systems that capture the whole spectrum are best suited for the identification of optimal wavebands and efficient algorithm development, which can be exploited in the development of spectral imaging systems with a limited number of wavebands (e.g., multispectral) systems that have the capability to meet the needs of real-time acquisition and processing. However, as noted previously, multispectral systems may not provide the degree of precision as hyperspectral systems.

## 3. Data Analysis and Modelling

The aim of using sensor technologies is to measure different grain traits rapidly and non-destructively based on specifications set by the needs of industry and grain trade standards. The challenge in the application of sensor technologies is that interpretation of the data can require complex algorithm development. Mathematical models are built to predict the desired trait or grain class using numerical or categorical descriptors or to discover patterns and relations in the data. The set of methodologies that 'learn' to build models from data (as opposed to being strictly programmed) are broadly referred to as machine learning methods. Machine learning is typically categorized based on the learning type (supervised or unsupervised) and learning models (regression, classification, clustering, dimensionality reduction, target detection).

At their core, machine learning algorithms are tasked with finding solutions to optimization problems. One of the simplest examples is ordinary least squares (OLS) regression, where parameters of a linear function are found that minimize the sum of the squares of the residuals (i.e., differences between observed variables and those predicted). More complex tasks, which involve high dimensional data and nonlinearities, require more complex algorithms, yet the basic principles (i.e., minimization of prediction error) remain the same.

This review provides brief descriptions of existing machine learning methods used for data analysis and modeling. Many of these techniques can be applied to the analysis of NIR spectral, digital RGB image, and multi/hyper-spectral imaging data and be used to perform both regression and classification of the trait of interest. Image data generally involves an additional set of unique methods using computer vision, a subset of machine learning which deals with vision-specific tasks that include image recognition, segmentation, target tracking, and motion estimation, among others.

### 3.1. Regression

The most basic form of regression is simple linear regression, in which a linear function is fitted to a predictor variable and a response variable. The linear equation consists of two unknown parameters (slope and intercept), which are found by minimizing the root-mean square (RMS) error of the resultant fit. RMS error is used in OLS, although alternatives such as mean absolute error (MAE) are sometimes used. The extension of simple linear regression to multiple predictor variables is known as Multiple Linear Regression (MLR). Another term, multivariate (or general) linear regression, refers to cases where there are both multiple predictors and multiple response variables. MLR is suited to modeling linear relationships with the use of one or more independent variables. A typical challenge in using regression to model spectral data is the high degree of multicollinearity (i.e., the correlation between predictor variables) in the data, which is usually remedied by using an independent subset of the data (e.g., Principal components) as predictor variables in

the model. Applications of MLR in Agriculture include the detection of fusarium head blight in wheat kernels [57], the determination of quality attributes of strawberries using hyperspectral imaging [85], and the assessment of lentil size traits [32].

### 3.2. Principal Component Analysis (PCA)

Principal component analysis (PCA) is one of the most widely used dimensionality reduction algorithms, typically used to represent high-dimensional correlated data. In general, a dataset may contain measurements of features (e.g., color, size, shape, etc.) from many samples. Principal components (PCs) refer to the axes of a new coordinate system along which the measurements are maximally correlated. The first PC accounts for the highest variance in the data, followed by the second, third, fourth, etc. Effectively, PCA allows high-dimensional data to be represented in fewer dimensions. PCA is often used to reduce highly correlated data to a small subset of independent variables, which can be used in other analysis techniques, such as regression. An extension to PCA known as principal component regression (PCR) uses the PCs of the explanatory variables to perform standard regression to describe a dependent variable. Application of PCA includes the study of the relation between dough properties and baking quality [89], characterization of desiccation of wheat kernels [55], and as a dimensionality reduction technique in many other applications [90].

### 3.3. Partial Least Squares (PLS)

Partial least squares or projection to latent structures (PLS) is a technique widely used in chemometrics. The principles of PLS are similar to PCR, i.e., the independent variables are first projected to a new space to reduce the dimensionality of the data and to infer a set of latent variables, which are then used to perform standard regression or classification. The technique is suitable for modeling highly collinear data, such as NIR spectra with broad chemical signatures that contain many redundancies. PLS is broadly used in chemometric analysis [54] for measuring key traits such as protein and moisture in grain [1,31,91,92].

### 3.4. Linear Discriminant Analysis (LDA)

Linear discriminant analysis is one of the standard methods used to classify linearly separable data into two or more categories [69]. The goal of LDA is to find a linear combination of predictors (i.e., projection to new feature space) that separates two or more classes in the data. The new feature space is obtained by finding a projection that maximizes the distance between the inter-class data while minimizing the intra-class data [93]. Application of LDA includes use in the differentiation of wheat classes with hyperspectral imaging [94], classification of defective vs. non-defective field peas using image features [13], and detection of fungal infection in pulses [95].

### 3.5. Support Vector Machine (SVM)

Support vector machines (SVMs) are methods used to classify data into two classes (binary classification). An SVM classifies data by finding the best hyperplane that separates all data points of one class from those of the other class. The best hyperplane in the case of SVMs is the one that creates the largest margin between the two classes. For multi-class problems, several binary problems must be applied. Other versions of SVMs include support vector regression (SVR) and least-squares support-vector machines (LS-SVMs).

SVMs are one of the most successful machine learning algorithms and have been widely used in industry and science, often providing results that are better than competing methods. Applications to agricultural sensor data modeling include use in the analysis of chalkiness in rice kernels [96], classification of contaminants in wheat [97] and quality grading of soybeans [98].

### 3.6. Decision Trees Learning

In decision trees or classification trees, the objective is to construct a flowchart for making decisions based on criteria related to a desired outcome. The decision trees are often constructed by experts with knowledge of the decision-making process. Decision tree learning (DTL) provides an algorithmic technique for constructing a tree based on data for use in the creation of predictive models for classification or regression. An extension to decision trees is known as Random Forests (RF), where many decision trees are constructed to provide a more robust framework and avoid the common overfitting pitfalls experienced in DTL. Overfitting typically occurs when the model has too many parameters compared to the number of predictors, which manifests as a good fit to the training data and poor prediction accuracy on new data. Overfitting is best managed using model validations, as discussed in Section 3.8. Applications of decision trees include discrimination of wheat varieties using image analysis [99] and classification of corn crop treatments from remote hyperspectral data [100].

### 3.7. Neural Networks and Deep Learning

Artificial neural networks (ANN) are a class of computation structures inspired by the functionality of the human brain. Basic units, referred to as artificial neurons, receive weighted inputs from other neurons and generate an output. The weight of a neural connection is typically normalized to the range 0 (i.e., no connection) and 1. Many neurons organized in networks enable complex information processing that includes pattern recognition, feature extraction, and learning. Training ANNs involve the adjustment of the weights of the network until the network output matches the desired output, given by a set of labeled training data (supervised learning). Many iterative optimization algorithms, such as stochastic gradient descent (SGD) and back-propagation, can be used to train ANNs.

Deep learning (DL) typically refers to ANNs with many layers of neurons. The different layers enable networks to learn complex data representations at multiple levels of abstraction. A common DL architecture is the convolution neural network (CNN), inspired by the neural connectivity in the visual system of mammals where connections are locally constrained. CNNs are particularly suited to image-related tasks. Other DL architectures include fully connected networks (FCN), recurrent neural networks (RNN), generative adversarial networks (GANs), and auto-encoder networks. Applications of ANNs include modeling the chemical constituency of cereals and pulses [31], classification of beans using images [101] and prediction of wheat yield [24], among many others [102].

### 3.8. Validation

One of the most common strategies used to assist in model selection and reduce overfitting is the use of cross-validation. During model development, random portions of the data are used to build the model, while the withheld portion is used to evaluate the performance of the model. There are many ways in which cross-validation can be performed, depending on what subset of the data is used for building the model (training) vs. validation. Popular strategies include *k*-fold cross-validation, repeated random subsampling validation, and leave-one-out cross-validation methods.

Independent validation is essential when assessing the feasibility of an application to test the accuracy and reliability of the algorithm(s), procedures and equipment. It is particularly important to determine if the calibration set is well constructed and can measure all variations in the desired trait. Ideally, the validation samples will be sourced from the new harvest, potentially a range of sites, covering all possible varieties of crops that will be assessed [103]. Independent validations and continuous monitoring are necessary when ensuring new sensor technologies are applied successfully, particularly for different growing locations, environmental conditions, transporting conditions and storage conditions such as those found on-farm.

## 4. Sensor Technologies Applied In-Line within Agricultural Systems

Real-time measurements of grain quality have been implemented in a range of agricultural processing industries on production lines to segregate between market quality and identify out-of-specification (or defect/contamination) products [29]. The advantage of such technologies is that they are non-destructive and can be applied with speed and accuracy, and do not tend to disrupt the processing of the commodity. Segregation technologies can be applied at multiple points within the processing chain, for example, measuring the bulk commodity before, during, and after processing, and have been applied in many food processing applications. Table 2 describes relevant applications for grains in food and agriculture. Various applications have been employed for a range of grains and seeds and industries to ensure a specific quality of end-product is achieved throughout processing (Table 2).

**Table 2.** Current sensor technologies applied to grains and seeds measured in-line for a range of quality parameters.

| Grain | Trait | Sensor | Data Analysis and Accuracy | Reference |
|---|---|---|---|---|
| **Lentil** | Seed length, mass, seed size index | Image analysis (multispectral), benchtop | MLR, image segmentation, $R^2$ for length 0.97, mass 0.96, seed size index 0.99 | [32] |
| **Pea** | Color, shape and size | Image analysis (multispectral), benchtop | Classification 77-87% in market grade classification | [13] |
| **Soybean** | Damaged kernels | Image analysis (visible), in-line benchtop | Neural Network classification 97.25% | [66] |
| **Wheat** | Protein, moisture | NIR, in-line mounted on a harvester | PLS regression, $R^2$ for protein 0.88, SECV 7.64; in-field protein vs. yield maps | [29,42] |
| **Rice** | Chalkiness | Image Analysis (RGB), benchtop | Classification (SVM) 89.9–90.3 2 sides 97.6–98.5% 3 sides | [96] |
| **Rice** | Classification of type (brown rice, buckwheat, barley, common rice, rough rice, | Image Analysis (visible), benchtop | Classification (PCA-NN) 90–100% classification of grain type within a sample | [104] |
| **Corn** | Broken, unthreshed, Material other than corn grain | Image analysis (RGB), mounted in-line on harvester | Color features, texture features, classification (SVM) Bivariate correlation 0.948 (with an $R^2 = 0.898$), based on weight percentages | [105] |
| **Coffee** | Sucrose, color (CIE L*, a*, b*) | Fourier-Transform NIR, mounted in-line | PLS regression, $R^2$ for sucrose 0.93 and color 0.85–0.94. The predicted data was used to monitor the roasting process. | [106] |

Full spectral or multispectral sensors have been successfully engineered within grain processing lines where the product is measured in-line, such as grain passing by a camera on a conveyor belt [32], or the grain is subsampled as it is harvested and measured in a sampling device with a NIR [42]. Once analyzed, the grain can be classified or quantified to inform decisions, such as to reject or modify the process when an out-of-specification limit is detected. For on-farm quality segregation, sensor technologies can be applied at-harvest as the grain is transferred into chaser bins or trucks (classifying the whole of load), or potentially the grain can be 'actively' segregated in-line using a diverting system as it is being loaded onto the truck. An example of 'active' segregating using a diverting system based on a binary decision is where high-protein wheat is separated from low-protein wheat into two bins [29]. It should be noted that the cost of the installation, measurements

representative of the batch (sampling), the accuracy of the algorithm and the speed of the decision-making process are major considerations in any in-line application.

Sensor technologies have been applied to coarse material, such as grains, with a range of accuracies (Table 2). The accuracies are dependent on many factors, some of these include the sensor-type, sample presentation when the measurement is taken, and the type of parameter measured (protein, moisture, grain size). In addition, if segregating the product into two streams, poor and good quality, then other considerations include processing speed and delay between the measurement and sorting device.

Successful applications (with an $R^2 > 0.88$ and classification models >77%) have been developed in the use of image analysis and near infrared technologies to classify and quantify grain based on color, shape, size, moisture, protein, broken, damaged and contaminated course samples. Key quality traits of major importance to the agricultural industry include protein, moisture, grain size, discolored or damaged grain, and contaminates (Table 2) could be objectively assessed with portable sensors on-farm. These traits could be incorporated into a decision process that assists growers and traders in managing the harvested grain based on quality traits to maximize profitability at the time of sale. The adoption of such technologies will be dependent on cost, the complexity of the application, ease of use and the likelihood of disruption or breakdown during use (Table 3).

**Table 3.** Complexity and cost of sensor technologies used in agriculture to predict grain quality traits.

| Sensor | Cost Estimation | Development and Equipment Complexity | Reference |
|---|---|---|---|
| NIR-spectra (in-laboratory) | USD 5000–50,000 | Standard software applications; used by industry; low maintenance; annual update of algorithms | [31,42,54] |
| Digital Camera (RGB) (in-laboratory) | USD 10,000–40,000 | Complex algorithm development; two-dimensional; low maintenance | [32,107,108] |
| Digital Camera (in-field) | USD 1000 to 10,000 | Complex algorithm development; two-dimensional; challenges validating at a single reading level; low maintenance | [109,110] |
| Stereo Camera | USD 20,000–30,000 | Complex algorithm development; multi-dimensional; low maintenance | [52] |
| Multi-spectral (in-laboratory) | USD 1000–40,000 | Complex algorithm development; image and spectral analysis; low maintenance; annual update of algorithms | [9,30,32,50] |
| Hyperspectral (in-laboratory) | USD 230,000 | Large data volumes, computation cost; complex algorithm development; challenges validating at a single seed level; research tool, image, and spectral analysis; low maintenance; annual update of algorithms | [14,60] |
| Hyperspectral (in-field) | USD 110,000–350,000 | Large data volumes, computation cost; complex algorithm development; challenges validating at a single reading level; used for predicting yield component, nitrogen inputs and grain protein; annual update of algorithms | [22,34,111] |
| Biospeckle (in-laboratory) | USD 15,000 | System not commercially available; complex algorithm development; low maintenance | [112–114] |

## 5. Considerations for Adoption

### 5.1. On-Farm Segregation and Storage

Segregation of grain, based on quality traits, can be implemented pre-harvest through the application of within-field zoning of different quality, during harvest using on-header sensing segregation, or post-harvest storage and out-loading. Growers can decide to implement these strategies individually or in combination. How a grower chooses to implement segregation will, to a large part, depend on the economics and logistics involved.

Cropping enterprises have historically used on-farm storage as a strategy to minimize economic risk and mitigate the impacts of variable market conditions. Grain storage provides options to growers at harvest by broadening the harvest window while also

preventing the loss of grain attributed to extreme weather events, crop shedding and/or lodging, as well as reducing the time trucks are unavailable due to unloading [115]. In addition, on-farm storage allows growers to temporarily hold grain until desired market prices can be received. In these instances, growers often blend grain of differing quality to reduce downgrading and segregate grain based on quality specifications [115].

While the economic benefits of segregating, blending and storing grain are evident, the cost of capital infrastructure requirements needs to be considered. One major issue associated with on-farm storage, particularly vertical systems, is the difficulty in determining the cost-benefit of increasing storage capacity, where fluctuating annual yields will often mean an underutilization of space or a requirement to store grain in temporary bunkers, use of grain bags or the decision to send directly to receival [115]. An added level of complexity with segregation is attempting to pre-determine the storage requirements for the potential grain yield and grade. Increasing the number of segregated or blended groups, a greater number of smaller storage vessels will be needed to decrease the size of each management unit and increase the opportunity to benefit from segregation. In addition to efficiencies gained by a general increase in on-farm storage, smaller and greater numbers of vertical storage systems will decrease the amount of grain requiring cleaning and increase the control growers will have to sell, feed out or store grain for future seeding.

### 5.2. Data and Interfaces

Developments in precision agriculture technologies have seen an increase in the intensity of information and data transfer required between sensors, computers, and users. In recent years, sensor capability, advanced computing, robotics, automation, wireless technology, GIS (Geographical Information Systems) and DBMS (Database Management Systems) have contributed to the complexity and rate of data communication [116,117]. While many efficiencies have been gained (i.e., Variable Rate Technology), the production of large data sets for growers and advisors to collect, decipher and utilize has often discouraged adoption. [118] identifies that this 'data overload' must be overcome by the development of data segregation tools, expert systems, and decision support capability. This 'data workflow' needs to be well designed such that user interaction is clear, unencumbered, and informative. This will require well-designed integration of hardware, software, and decision support tools irrespective of how the grower chooses to implement segregation.

Integration of data systems is an issue continually raised in the agricultural technology sector. Surveyed growers, e.g., [117], have expressed a preference for interface systems that are either independent but intuitive to operate or, preferably, integrate into current systems and monitors. However, despite the generation of standardized data protocols for farm technology, i.e., ADIS—Agricultural Data Interchange Standard and CANbus [116], many manufacturers of agricultural equipment have restricted data to proprietary formats that limit their interoperability. Adoption of a segregation system would likely be influenced by the ability of the technology to integrate with existing systems both for ease of operation and centralizing farm data. Advancements in telecommunication networks, such as the Internet of Things (IoT) and wireless networks, have significantly increased the opportunities for connectivity and integration of on-farm sensors [119]. Despite this, there remains a need for new interfaces to be as intuitive, reliable and available as possible to promote adoption and satisfy the needs of growers [120]. Finally, the intellectual property (IP) around who holds and has access to data and data governance is an issue that has become increasingly important in the use of these data-intensive technologies, and there are national and international regulations that are being developed [121].

### 5.3. Calibration and Technical Support

A common feature of sensor-based tools is the requirement for regular calibration and correction. Most sensors, such as multispectral imaging systems, require frequent radiometric calibrations and corrections, which are critical to obtaining data that can be compared over multiple time periods [122] by accounting for changes in environmental

conditions such as light intensity and atmospheric and surface conditions [123]. A sensor-based system, whether pre-, at or post-harvest, will inherently require calibration and correction to account for changes in crop species, variety, location, time, and additional variation attributable to the sensor type and measurement parameters. For growers, the perceived lack of benefit of adding this level of complexity to a system is often what drives growers to favor other emerging technologies (i.e., disease/herbicide tolerant varieties) and limit sensor adoption [124]. A study by [125] summarized over 30 grower surveys, indicated that in addition to calibration and correction, troubleshooting, training and support for new farm technology also determined a grower's PEA 'Perceived Ease of Use', and that PEA was positively related to adoption. Thus, along with the development of robust hardware, software, interfaces and data workflows, competent technical support is required for these technologies to be adopted by growers.

## 6. Gaps, Challenges, and Benefits

Current systems that classify grain quality to determine graded segregations for trade require a range of standardized methodologies and instrumentation (including benchtop-NIR, a range of nested sieves, balances, chondrometers, visual grain inspection, etc.). Other considerations for the grower when classifying grain quality on-farm include training operators each season, ensuring the sensor windows are clean and free of dust, the influence of temperatures on instrumentation, adequate storage of instrumentation, ways of recording data and relating the information back to each load, chaser bin or storage facility, and ensuring that the instrumentation is kept clean (free of mice damage and dust, etc.). Clearly, data workflows that are as automated as possible would ease the ability of growers to adopt sensor technology. The benefits and challenges of applying sensors that can classify grain quality traits on-farm are discussed.

Developing sensor technology systems designed to measure a range of objective and subjective traits, such as grain size and visually assessed traits, would be advantageous to growers, particularly if the devices incorporated portable Wi-Fi or Lorawan and were permanently installed on grain-moving systems (i.e., harvesters, augers, belt movers). Having a system that simultaneously measures multiple traits improves efficiencies and synergies as one system would replace many tests (sieves, balances, NIR, chondrometer, visual inspection). This includes the value in synergies of having fewer instruments to do more on-farm and the associated benefits to growers.

The ability to actively manage grain throughout production, storage and transport requires the development of systems that can maximize growers' returns. As grain market classes are defined by specific varieties and grain traits important for end-users, the development of applications (sensors and algorithms) on-farm will enable growers to objectively quantify the value of their crop. Key quality traits for growers include protein and moisture percentage, grain size and defects such as weather damaged, stained grain, and contamination. Currently, there are instruments with the ability to test both small parcels of grain (~200 g), large scale (in-line harvester or auger, representing ~100–500 tonnes), and portable sensors that measure crop canopy for crop growth and yield components. Through the development of grain-quality applications to quantify market class, growers will be able to segregate grain on-farm to potentially achieve higher aggregate prices and capture premiums for higher grades and minimize downgrades.

The use of on-farm storage is increasing each year and is altering harvest logistics. With the addition of on-farm storage to a farming enterprise, growers can delay strategic decisions for post-harvest delivery to grain agents. Delaying delivery minimizes exposure to price fluctuations as traditionally, growers would off-load the grain directly after harvest when grain supplies are high and therefore driving grain prices down. Storage facilities can also allow growers to focus on harvesting their crops and decrease the time taken to off-load and transport grain during peak times. However, there is a risk associated with on-farm storage, as sub-optimal storage practices can result in spoilage of grain and a loss

in quality. Grain storage risks could be potentially minimized if sensor technologies were used to monitor grain quality variation over time.

Grain is increasingly being utilized by non-traditional markets, for example, plant-based proteins, and there is a need to identify the quality requirements so that processing needs are met for these emerging and expanding opportunities. The demand for protein-rich meat alternatives is increasing, and the plant-based protein industry is estimated to reach a value of USD 17.8 billion dollars globally by 2027 [126]. Therefore, future considerations for the grain sector include a range of sensor tools that identify the quality specifications needed to produce a consistent high-value product. Developing sensor technologies that can capture these opportunities will ensure that the grains industry sector remains competitive as these markets emerge.

Another opportunity for the application of sensors on-farm is to meet the increasing demand by consumers to know more about the provenance of food, its composition, quality and whether it adheres to ethical considerations. Therefore, growers will require tools that can link crop data and trace the source of production, identify quality components of the product, and other food system characteristics. These tools and data have to potential to; inform systems that minimize waste, improve operational efficiencies and processing, assist in demand forecasting, manage compliance and quality issues detected along the supply chain, and evaluate the production system from harvesting to off-loading post-farm gate. The development of innovative e-tools that accurately record on-farm analytics, link source (seed and location), the authenticity of quality (variety, composition), storage and handling conditions can ultimately be fed into the supply chain and utilized by growers, grain agents, food processors and consumers who can determine product value, build product confidence, and demonstrate evidence of quality product from paddock to plate.

In the future, sensor technologies integrated at different stages of the grain-value chain will offer practical solutions that enable growers and the grains sector to identify, manage and segregate grain to maximize product quality and value.

**Author Contributions:** Conceptualization, C.K.W., J.F.P., J.G.N.; writing—original draft preparation, C.K.W., S.A., A.J.W., A.J.D., J.G.N., A.B.C., G.J.F., J.F.P.; writing—review and editing, C.K.W., S.A., A.J.W., A.J.D., J.G.N., G.J.F., J.F.P., L.S.M.; visualization, C.K.W.; supervision, C.K.W., J.F.P., J.G.N., G.J.F.; project administration, C.K.W.; funding acquisition, J.F.P., J.G.N. All authors have read and agreed to the published version of the manuscript.

**Funding:** This research was funded through the Victorian Grains Innovation Partnership, a collaboration between Agriculture Victoria (Ag Vic) and the Australian Grains Research Development Corporation (GRDC). Project VGIP1C-IDGRDC8049295. On Farm Grain Quality Capture.

**Data Availability Statement:** No new data were created or analyzed in this study. Data sharing is not applicable to this article.

**Conflicts of Interest:** The authors declare no conflict of interest.

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
