# Peer review of "Technologies and Data Analytics to Manage Grain Quality On-Farm—A Review"

_agronomy, doi:10.3390/agronomy13041129_

Round 1

Reviewer 1 Report

Extensive review focused on up-to -date topic. I am not sure if it is good to combine both instrumental methods and data analysis and modelling in just one review. For me it would be better to have a chapter focused on economics of systems mentioned.

Table 1  - It is interrupted within one page. Would be better to include lines dividing table into sections of rows belonging to each sensor type – in actual version it is not always clear which row belongs to which type of equipment.  Multispectral (benchtop) is named twice in Tab 1.

Author Response

Not many reviews have been undertaken summarizing sensors that can be utilized on-farm to measure quality.  Many have been undertaken on predicting yield, biomass, N-utilization and disease.  However, few have focussed on managing quality on-farm.  The authors felt it important to summarize not only the sensors but the data analysis approaches to highlight differences in measuring quality to other important agronomic abiotic and biotic factors that affect crop production. However, reviewer 1 has given the authors a concept to write another manuscript on the importance of the economics of different systems that influences industry uptake.

The authors checked through the manuscript and undertook a spellcheck and the manuscript is updated with US spelling.  

The authors corrected the formatting of Table 1 and is now no longer interrupted.  And duplicated headings have been removed. Lines have been added in between the sections of Table 1.

The authors appreciate the reviewer's comments and time taken to undertake the review. 

Reviewer 2 Report

This is a very important review paper summarising current sensor technologies that are in the research phase and can potentially be applied on-farm within the next five to ten years, algorithm approaches that can be applied, and the planning, logistics and decision-making processes involved with determining grain quality on-farm to classify market value. Because people are trying their best to promote the application of sensor technologies expansion and adoption on-farm, which would make it possible that growers can identify the impact and manage the harvesting operation to meet a range of quality targets and provide an economic advantage to the farming enterprise, although this conception needs time to realize.

1. The main question addressed by the research is relevant in the field.

2. The paper added new knowledge to the subject area compared with other published papers.

3. The conclusions are consistent with the evidence and arguments presented, and they addressed the main question posed.

4. The references are appropriate.

5. It is not necessary to split Table 1 into two parts.

6. Table 2 is not needed, since the models and algorithms all were introduced in detail in the text.

7. Table 3 can also be omitted, if the Statistical Measures and the Abbreviations are necessary, they can be written in the text.

8. Table 6 can be deleted, as the perceived benefits and challenges in applying sensor technologies were already introduced in the text.

Suggestions:

Some parts such as “1 introduction, 2.1, 2.3 and 2.4, might be condensed by at least one fourth of the words

Page 10, “2.4. Hyperspectral should be “2.5 Hyperspectral”, see page 9 “2.4. Multispectral Imaging”.

Author Response

Table 1 has been merged into one format. Table 2, Table 3 and Table 6 have been removed.  It is not necessary to duplicate what is included in the text. The headings on Page 10 have been updated from 2.4 to 2.5 for the Hyperspectral section.

The authors appreciate the recommendation to condense the introduction (1) and sensor technologies (2) sections, however the authors felt it important to highlight the areas that sensors can be applied on-farm in the introduction (section 1) and the sensors/applications that have been developed to date (section 2).  These areas were seen to be distinct by the authors and this is why the paper was structured as so. 

The authors appreciate the comments and suggestions made by this reviewer.

Reviewer 3 Report

This review paper is very interesting and meaningful. There is a specific suggestion for further improvement.

With the development of technology, more and more sensors were mounted on the satellites in recent years, and the satellite data, with various spatial and spectral resolutions, were widely used by numerous researchers for crop growth condition monitoring and crop quality evaluation, i.e. precision agriculture. However, relative little summary and discussion on this regard have been described in this manuscript. I suggest to add the relevant content in the former reviews and the latter challenges, to make the paper better.

Author Response

This paper focusses on on-farm applications and current findings.  I agree with satellites there is now another area that can be developed in terms of determining quality in-paddock.  To include this in the review would add another 5-10 pages.  Therefore we feel it is best to be addressed in another review.  There are many available in remote sensing, particularly covering grain yield, disease and it is a rapidly moving research space.  The authors are currently developing another review and research manuscript covering this very topic with quality as the focus.  Therefore, to keep this manuscript succinct we will not include the spatial and temporal spectral research that has been undertaken to date in this review.

Reviewer 4 Report

Dear Authors,

Thank you!

Your paper is very interesting and topical because of the problems of export of Ukraine grain due to Russian War.

Your paper is good and well-written. However, before publishing - I think that it needs some small revision:

- Lines 47-48: Do we need some reference here?

- Lines 52-56: Missing reference?

- Line 133: I propose to add a new heading:

2. Options for on-farm sensoring

2.1 Application of sensing grain guality...

- Table 1. Please improve the lay-out of table.

- Line 315. Sfm >> SfM

- Line 348-358. Missing references?

- Table 2. Place of table? Please place it after when you mention it in the text of your paper.

- Tables 4 and 5. Please improve the layout of the tables.

- Chapter 6. Did I miss one thing? I kindly call for you analyse briefly what is the effect of the grain market price on the willingness of the growers to study the quality of grain. 

Author Response

- Lines 47-48: Referenced, refer to line 48.

- Lines 52-56: Referenced, refer to line 57.

- Line 133: Head altered to the proposed title "Options for on-farm sensing grain quality"

- Table 1. Layout improved as recommended by reviewer 1

- Line 315. Sfm updated to SfM as suggested by reviewer

- Line 348-358. Referenced at line 351.

- Table 2. Table removed, information included in text therefore redundant.

- Tables 4 and 5. Tables renumbered to Table 2 and 3.  Columns made wider but would be better formatted to be landscape rather than portrait.

- Chapter 6. Reviewer comment "Did I miss one thing? I kindly call for you analyse briefly what is the effect of the grain market price on the willingness of the growers to study the quality of grain."  Grain quality is the equivalent of market price.  The grain size, physical appearance, % defects, damaged grain, protein content are quality traits that are used at receival sites for determining the grain price.  Therefore grain market price is determined by grain quality, if growers could determine grain quality with sensors on-farm prior to sale then they would be able to minimize risk by knowing the value of their product prior to delivery.  However currently growers do not have the technology to classify the harvested product on-farm without considerable labour and training. This is the point of Chapter 6, to outline the Gaps, Challenges and Benefits.

Thank you for your comments to improve the manuscript.
